# Towards Mitigating Systematics in Large-Scale Surveys via Few-Shot Optimal Transport-Based Feature Alignment

**Sultan Hassan**
Space Telescope Science Institute
sultanier@gmail.com

**Sambatra Andrianomena**
South African Radio Astronomy Observatory
University of the Western Cape
andrianomena@gmail.com

**Benjamin D. Wandelt**
Johns Hopkins University
wandelt@jhu.edu

## Abstract

Systematics contaminate observables, leading to distribution shifts relative to theoretically simulated signals—posing a major challenge for using pre-trained models to label such observables. Since systematics are often poorly understood and difficult to model, removing them directly and entirely may not be feasible. To address this challenge, we propose a novel method that aligns learned features between in-distribution (ID) and out-of-distribution (OOD) samples by optimizing a feature-alignment loss on the representations extracted from a pre-trained ID model. We first experimentally validate the method on the MNIST dataset using possible alignment losses, including mean squared error and optimal transport, and subsequently apply it to large-scale maps of neutral hydrogen. Our results show that optimal transport is particularly effective at aligning OOD features when parity between ID and OOD samples is unknown, even with limited data—mimicking real-world conditions in extracting information from large-scale surveys. Our code is available at ⊙.

## 1 Introduction

Upcoming large-scale facilities, such as the Square Kilometer Array (SKA, Mellema et al., 2013), the Vera C. Rubin Observatory Legacy Survey of Space and Time (LSST, Ivezić et al., 2019), Nancy Grace Roman Space Telescope (Roman, Spergel et al., 2015), Spectro-Photometer for the History of the Universe, Epoch of Reionization, and Ices Explorer (SPHEREx, Doré et al., 2014) and Euclid (Racca et al., 2016), will enable imaging the neutral hydrogen (HI) distribution in the early universe with unprecedented sensitivity over cosmological volumes. A key challenge in extracting cosmological and astrophysical information from these surveys lies in our ability to mitigate systematics, which introduce significant distribution shifts relative to theoretically simulated signals.

However, systematics are often poorly understood and challenging to model, and thus removing them entirely may not be feasible. As a result, many observables may exhibit distribution shifts and lie out of distribution relative to the simulated signal. Furthermore, theoretical and numerical models remain far from complete due to limited computational capabilities, which constrain our ability to fully capture the multiscale nature of astrophysics and cosmology—from the interstellar, through the circumgalactic, to the intergalactic medium. Consequently, it is often the case that observations

and theory do not arise from the same underlying distribution. This suggests that domain adaptation techniques may need to be consistently incorporated when extracting information from observations.

Optimal Transport (OT) has become a powerful tool for unsupervised and semi-supervised domain adaptation, offering a principled way to align data distributions without requiring exact sample correspondences. For example, Courty et al. (2017) introduced a joint distribution OT framework for deep ranking and classification tasks, demonstrating state-of-the-art results on benchmark datasets by minimizing a coupling between source and target distributions. More recently, Andrianomena and Hassan (2025) employed an adversarial approach combined with OT to label HI maps from one simulation using models pre-trained on another, highlighting OT's utility in bridging simulation gaps in astrophysical applications.

## 2 Method

We consider the following setup: a large set of in-distribution (ID) simulated signals ($x_{\mathrm{ID}}$) with labels ($y_{\mathrm{ID}}$), and a small set of noise-contaminated out-of-distribution (OOD) observables ($x_{\mathrm{OOD}}$) with unknown labels. Our goal is to infer the OOD labels by leveraging the ID data.

We first pre-train a model on the ID samples to learn the mapping $f : x_{\mathrm{ID}} \to y_{\mathrm{ID}}$. Since $y_{\mathrm{OOD}}$ is unavailable, standard transfer learning cannot be applied, as it requires labeled target data for adaptation. The only information available from the OOD data is the set of representations produced by the pre-trained model.

To enable labeling, we align the OOD representations to the ID representations. These representations are obtained from the pre-trained model. We compare two feature-alignment losses—MSE and Optimal Transport (OT)—to assess which better reduces the discrepancy between the two representation sets. The procedure consists of:

1. **Pre-training:** Train $f$ on ID data and fix this model as the reference for ID representations.
2. **Feature extraction:** Create two identical copies of the pre-trained model: one frozen for extracting ID features, and one trainable for extracting and updating OOD features.
3. **Alignment:** Optimize the trainable model by minimizing an alignment loss between the two feature sets. We track downstream label-recovery performance to compare losses.

The schematic and corresponding algorithm are shown in Figure 1 and Algorithm 1.

**Feature Alignment Losses: MSE vs. Optimal Transport.** Given ID features $\{z_i^{\mathrm{ID}}\}_{i=1}^N$ and OOD features $\{z_j^{\mathrm{OOD}}\}_{j=1}^N$, extracted from all $N$ layers of the model, we seek to align their distributions by minimizing an appropriate distance metric.

A common choice is the **Mean Squared Error (MSE)**, which assumes a fixed one-to-one correspondence and minimizes $\mathcal{L}_{\mathrm{MSE}} = \frac{1}{N} \sum_i |z_i^{\mathrm{ID}} - z_i^{\mathrm{OOD}}|_2^2$. This works only when paired samples exist; in unpaired settings, it forces incorrect matches and can hinder alignment.

To address the limitations of MSE in unpaired settings, we use **Optimal Transport (OT)** to align the distributions of ID and OOD features. Intuitively, OT finds a "soft matching" between the two sets of features by minimizing the overall cost of moving the OOD features to resemble the ID features.

Formally, given ID features $\{z_i^{\mathrm{ID}}\}$ and OOD features $\{z_j^{\mathrm{OOD}}\}$, OT solves for a transport plan $\gamma \in \mathbb{R}^{N \times M}$ that minimizes

$$\mathcal{L}_{\mathrm{OT}} = \min_{\gamma \in \Pi(\mu, \nu)} \sum_{i,j} \gamma_{i,j}\, c(z_i^{\mathrm{ID}}, z_j^{\mathrm{OOD}}),$$

where $c(\cdot, \cdot)$ is a ground cost (here, squared Euclidean distance), and $\Pi(\mu, \nu)$ is the set of all transport plans that respect the ID and OOD feature distributions. Unlike MSE, OT does **not require a one-to-one correspondence** between features, making it well-suited for aligning unpaired or few-shot OOD data. For the OT implementation, we tested two libraries, GeomLoss (Feydy et al., 2019) and POT (Flamary et al., 2024, 2021), and obtained similar results. We then compare OT's effectiveness against MSE by evaluating how well it improves downstream label prediction.

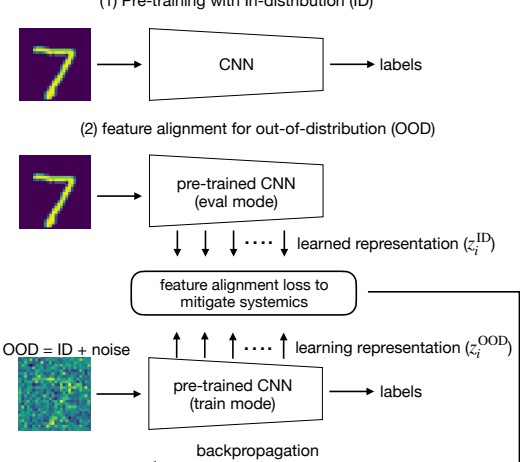

(1) Pre-training with In-distribution (ID)

CNN → labels

(2) feature alignment for out-of-distribution (OOD)

pre-trained CNN (eval mode)

↓ ↓ ↓ ⋯ ↓ learned representation ($z_i^{\mathrm{ID}}$)

feature alignment loss to mitigate systemics

OOD = ID + noise

↑ ↑ ↑ ⋯ ↑ learning representation ($z_i^{\mathrm{OOD}}$)

pre-trained CNN (train mode) → labels

backpropagation

Figure 1: Schematic view of the steps used to mitigate the impact of systematics on labeling observables. The first phase involves pre-training on ID samples, followed by a second phase where the pre-trained model is optimized using a feature alignment loss between ID and OOD samples.

**Algorithm 1** Feature Alignment for OOD labeling

1: **Input:** ID samples $(x_{\mathrm{ID}}, y_{\mathrm{ID}})$, OOD samples $x_{\mathrm{OOD}}$
2: **Output:** Predicted OOD labels $\hat{y}_{\mathrm{OOD}}$
3: **Step 1: Pre-training**
4: Train $f$ on $(x_{\mathrm{ID}}, y_{\mathrm{ID}})$ to learn $x_{\mathrm{ID}} \to y_{\mathrm{ID}}$
5: **Step 2: Feature Extraction**
6: Create two copies of $f$: $f_{\mathrm{ID}}$ (frozen) and $f_{\mathrm{OOD}}$ (trainable)
7: Extract features $z^{\mathrm{ID}} = f_{\mathrm{ID}}(x_{\mathrm{ID}})$ and $z^{\mathrm{OOD}} = f_{\mathrm{OOD}}(x_{\mathrm{OOD}})$
8: **Step 3: Feature Alignment**
9: Choose alignment loss $\mathcal{L}_{\mathrm{align}}$: MSE or OT
10: **while** not converged **do**
11:   Update $f_{\mathrm{OOD}}$ to minimize $\mathcal{L}_{\mathrm{align}}$
12:   Re-extract OOD features $z^{\mathrm{OOD}}$
13: **end while**
14: **Step 4: OOD Label Prediction**
15: $\hat{y}_{\mathrm{OOD}} = f_{\mathrm{OOD}}(x_{\mathrm{OOD}})$

## 3 Experimental Validation on MNIST

To validate our method, we use the MNIST dataset for simplicity. To simulate out-of-distribution (OOD) data, we add independent white noise with a standard deviation of 0.7 to each MNIST image. This perturbation breaks spatial correlations and introduces high-frequency modes, emulating realistic systematic contamination. In the pre-training phase, we adopt a simple convolutional neural network (CNN) consisting of two convolutional layers followed by two fully connected layers. The model is trained using the Adam optimizer with a cross-entropy loss function and a learning rate of $10^{-3}$ for 10 epochs, achieving an accuracy of 0.985 on the ID test samples.

We then evaluate our method under three scenarios of increasing difficulty, designed to reflect progressively more realistic conditions encountered when labeling OOD observables during feature alignment:

- Ideal Alignment (Paired / Many Samples): We assume that a one-to-one pairing between in-distribution (ID) and OOD samples is known, and a large number of OOD realizations are available. This setting represents the best-case scenario for alignment. Many samples refers to the full MNIST dataset of 60,000 images for training and 10,000 for testing.

- Unpaired Alignment (Unpaired / Many Samples): We assume no known correspondence between ID and OOD samples, although a large number of OOD realizations are still available. This setting reflects the common case where sample pairing is unknown. This unpairing is achieved by randomly shuffling the data.

- Few-Shot Unpaired Alignment (Unpaired / Few Samples): We assume no known pairing and only a limited number of OOD samples. This is the most realistic and challenging scenario, where data is sparse and pairing is unavailable. Few samples refers to using a randomly selected subset of 32 images.

These three setups allow us to systematically assess how the availability of data and pairing information affects the success of feature alignment using different loss functions.

Figure 2 shows the classification accuracy evolution during feature alignment using MSE and OT for OOD samples. All scenarios start from the accuracy of the pre-trained model on OODs which is 0.5. Significant improvement is achieved after a single backpropagation step in the many-sample case with OT, yielding an accuracy gain of approximately 40% (from 0.5 to 0.9) and 30% (from 0.5 to 0.8)

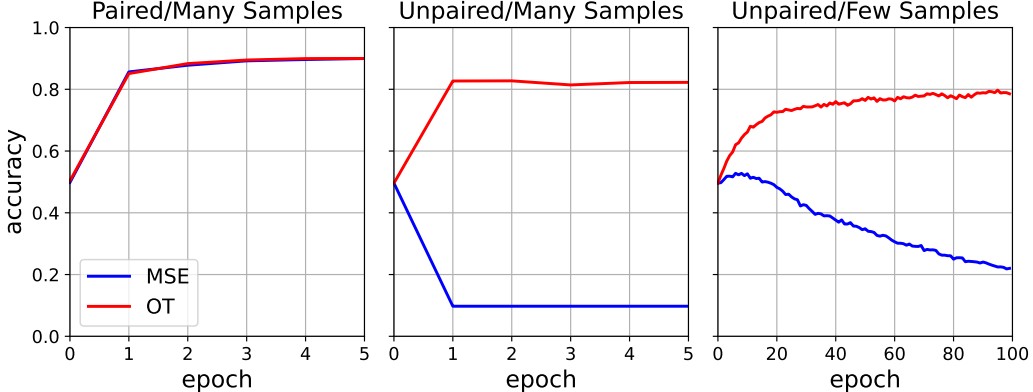

Figure 2: OOD classification accuracy evolution after feature alignment using mean squared error (MSE) and optimal transport (OT). At the beginning of the feature alignment training, the pre-trained model has very high accuracy on the IDs (0.985) but only random-guess–level performance on the OODs (0.5). This OOD accuracy of 0.5 serves as the starting point at epoch 0. Significant improvement is achieved after a single backpropagation step in the many-sample case with OT, whereas much more optimization steps ($\sim$ 20-40) are required in the few-sample case. In all cases, feature alignment with OT improves OOD accuracy without using any labels.

for the paired and unpaired cases, respectively. In the case of fewer unpaired samples, significantly more optimization steps ($\sim$ 40) are required to achieve a comparable accuracy gain of $\sim$ 30%, as seen in the many unpaired sample setting. In all cases, we find comparable accuracy on the testing and validation sets. This exercise shows that feature alignment with OT improves OOD accuracy without using any labels.

However, the Mean squared error (MSE) performs pointwise alignment, assuming a one-to-one correspondence between features of in-distribution (ID) and out-of-distribution (OOD) samples. It works well when such pairings are known (i.e., in the paired setting), but fails in the unpaired case, where sample indices do not correspond—resulting in spurious or incorrect alignment.

In contrast, optimal transport (OT) aligns entire feature distributions rather than individual samples. It computes a transport plan that softly matches OOD features to ID features in a globally optimal way, without requiring explicit pairing. This allows OT to handle both paired and unpaired scenarios effectively, including settings with few OOD samples.

## 4 Estimating $\Omega_m$ using Aligned Features on HI Intensity Maps

We now apply the OT-based feature alignment strategy to HI intensity maps from the CAMELS dataset (Villaescusa-Navarro et al., 2022). We use the publicly available maps from the CAMELS Multifield dataset at redshift $z = 0$, specifically from the Latin Hypercube (LH) set, which contains 15,000 images spanning 1,000 cosmological and astrophysical parameter combinations. We focus on estimating the matter density parameter $\Omega_m$ to evaluate whether the method performs effectively in a scenario closer to real observations.

To create the OOD samples, we mimic the addition of thermal noise encountered in radio interferometric observations by injecting white noise with a standard deviation of 0.7 (see Figure 3), after standardizing the data to have zero mean and unit standard deviation. For the pre-training phase, we use a simple convolutional neural network (CNN) consisting of three convolutional layers followed by four fully connected layers to learn the mapping from HI maps to $\Omega_m$. For reproducibility, we train the model using the mean squared error (MSE) loss with the ADAM optimizer, a learning rate of $10^{-3}$, and a total of 100 epochs. We randomly select 12,000 images for training, and 3,000 images for testing and validation. The pre-trained model achieves a coefficient of determination $R^2 = 0.92$ on the ID test set but $R^2 = -5.3$ on the OOD test set. All experiments were run with fixed random seeds for model initialization and data shuffling to ensure consistent results across runs.

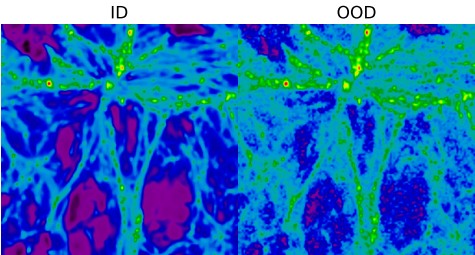

Figure 3: A random realization of a $256 \times 256$ HI map at redshift $z = 0$ from the CAMELS dataset. The out-of-distribution (OOD) sample is generated by adding white noise, mimicking the effect of thermal noise in radio interferometric observations.

| Alignment Scenario | $R^2$ | Optimization steps |
|---|---|---|
| Paired/Many Samples | -5.3 → 0.86 | 1 |
| Unpaired/Many Samples | -5.3 → 0.82 | 1 |
| Unpaired/Few Samples | -5.3 → 0.86 | 50 |

Table 1: OOD accuracy improvement using OT feature alignment across different scenarios to infer $\Omega_{\mathrm{m}}$, reported as a function of optimization steps. For reference, the in-distribution (ID) testing set achieves $R^2 = 0.92$.

We then report the alignment results as a function of optimization steps for each scenario in Table 1. Similar to the MNIST case in the previous section, significant improvement occurs after a single optimization step when many samples are available. In contrast, unpaired few-shot alignment requires more optimization steps to achieve comparable results.

Although the *Unpaired / Many Samples* scenario provides more data, the larger sample size may introduce greater variability and noise in the empirical feature distribution, which can make the optimal transport (OT) alignment less precise or harder to converge. In contrast, the *Unpaired / Few Samples* case, despite using less data, might benefit from a cleaner and more concentrated feature distribution, enabling OT to find a more stable and effective alignment path. This counterintuitive behavior highlights the importance of sample quality and feature structure in OT-based domain adaptation.

## 5 Conclusion and Future work

In this work, we have demonstrated that optimal transport can effectively enable few-shot feature alignment, offering a promising approach to mitigating systematics in large-scale surveys. This method paves the way for more robust and accurate inference in scenarios with limited labeled data and significant domain shifts.

Future work will focus on extending this framework to incorporate uncertainty quantification and scalability study, enabling more reliable inference under domain shifts for expected big datasets from upcoming surveys. Additionally, extending the comparison beyond MSE to include other methods such as adversarial, prototype-based matching, contrastive learning, and normalization-based approaches could further validate the effectiveness of optimal transport alignment, especially in highly complex and heterogeneous data environments. We will also consider incorporating more realistic noise realizations to better forecast conditions for upcoming experiments. Moreover, we plan to apply this method to generative architectures, such as diffusion models or GANs, to perform feature alignment in these contexts.

## 6 Broader impact

Our proposed feature alignment method using Optimal Transport (OT) has broad applicability beyond astrophysical data. By aligning learned representations between in-distribution and out-of-distribution samples, it provides a robust framework for domain adaptation in scenarios where labeled OOD data is scarce or unavailable. To ensure reproducibility in other domains, one can apply the same pipeline: extract features from a pre-trained model, compute the OT transport plan using a suitable cost function (e.g., squared Euclidean distance), and update the OOD representations via the chosen optimizer. This approach can enhance the reliability of machine learning models in diverse scientific fields, including medical imaging and remote sensing, where systematic shifts often challenge model generalization.

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
