# OpenReview forum: "Towards Mitigating Systematics in Large-Scale Surveys via Few-Shot Optimal Transport-Based Feature Alignment"
_NeurIPS.cc/2025/Workshop/UniReps — UniReps2025_

### Official Review · Reviewer_ZJDw · 2025-09-07
**Towards Mitigating Systematics in Large-Scale Surveys via Few-Shot Optimal Transport-Based Feature Alignment**

**Confidence:** 4

**Review:**

The paper addresses challenges caused by systematics in large-scale astrophysical surveys. The authors propose using optimal transport to align features between in-distribution and out-of-distribution data. Experiments on MNIST and HI intensity maps show that this method improves labeling accuracy, even in unpaired or few-sample scenarios. The approach is original, technically sound, and relevant to the UniReps community.

**Strengths**
* The paper identifies how systematics can shift distributions in large-scale surveys and demonstrates a method to correct them.
* The experiments include both paired and unpaired scenarios, as well as cases with very few out-of-distribution samples, showing practical relevance.
* Optimal transport achieves better feature alignment than mean squared error, as shown in MNIST and HI map experiments.
* The authors apply the method to HI intensity maps and successfully estimate Ωm, illustrating its real-world value.
* Quantitative results, including accuracy and R², are reported clearly and support the conclusions.
* The discussion explains how sample quality and feature distribution influence the effectiveness of alignment.

**Weaknesses**
* The paper does not compare the proposed method to other domain adaptation techniques, such as adversarial or normalization-based approaches.
* The computational cost and scalability of optimal transport for larger datasets are not addressed.
* There are some sections of the methodology that are difficult to follow for readers without prior experience with optimal transport.
* The broader impact and reproducibility are mentioned briefly but could be expanded for clarity.

**Suggestions**
* The authors should include comparisons to other domain adaptation methods on HI maps to provide context for the improvements achieved.
* They can provide information on computational cost and how it scales with dataset size would strengthen the practical evaluation.
* Step-by-step explanations or pseudocode could make the methodology easier to follow and reproduce.
* Also, including a discussion of uncertainty quantification and potential limitations in different scientific domains would enhance the impact.

The paper presents a meaningful contribution to feature alignment and systematics mitigation in large-scale surveys. The experiments and results are strong and demonstrate real-world applicability. Despite some gaps in comparison and methodological clarity, the work is well-executed, technically sound, and relevant. I recommend acceptance.

**Score:**

4

**Topic Fit:**

2

---

### Official Review · Reviewer_DPTQ · 2025-09-15

**Confidence:** 3

**Review:**

This work aims to address the issue that systematic errors lead to distribution shifts relative to theoretical results used in pre-trained models for labeling observables, thereby affecting the performance of these models. This paper proposes feature alignment using a pair of models, one trained on in-distribution and the other then adjusted using out-of-distribution samples with a loss defined as Optimal Transport. Then, it validates it by comparing losses using MSE and Optimal Transport.


## Strengths

It states relevant problem and domain examples.

Provides a reasonable method and related work.

Provides a set of examples that test the proposed methodology in setups that can help understand its effects.

## Weaknesses

From the results, it seems that the reduction in accuracy could be caused by all the noise in the OOD samples, as shown in the Unpaired Alignment (Unpaired / Many Samples), and it seems to improve as the fewer OOD samples used to train

It would be useful to assess the accuracy of the pretrained model on OOD and ID data without updating it with the proposed method, to determine if the proposed method truly enhances performance; otherwise, there is no strong evidence to support the claim.

It would be beneficial to clarify the validation and test datasets and to present the results for each to demonstrate the method's actual impact.

## Clarity

A bit more clarity in the explanation of each part of the method could help the reader distinguish which part is being compared to what. Similarly, the results might need a bit more detail to clarify what the numbers refer to. For example number of samples, training, validation, and test sets. Otherwise, it might give the impression of being too ambiguous.

Comments:

The abstract does not describe the proposed method; it might be helpful to provide a brief description before mentioning what was done to validate it.

The term systematics might be niche, so if it is expected not to be common jargon in the venue, a brief explanation of the term might help.

It is mentioned in line 53 that conventional transfer learning techniques for few-shot training on OOD data are not feasible; a brief explanation of why might help strengthen the proposed idea.

I hope this feedback is useful.

Good luck.



## Overall Recommendation
- **Weak Accept**

**Score:**

3

**Topic Fit:**

2

---

### Official Review · Reviewer_qTL5 · 2025-09-17
**A clear well structured paper**

**Confidence:** 4

**Review:**

Strengths
- The paper proposes a novel application of Optimal Transport, which is particularly effective in few-shot, unpaired, and unlabeled data scenarios
- The paper is clear and well-structured, with a distinct pre-training phase and feature alignment phase outlined in the Method section. Particularly the use of Figure 1 to provide a clear schematic overview of the two-phase method, which is a key strength for understanding the proposed approach.
-  The authors provide a clear, step-by-step experimental validation on the MNIST dataset under various distinct scenarios of increasing difficulty.
- The paper successfully applies the method to realistic astronomical problem, highlighting it's importance in real-world scenarios

Weakness
- The experiments are limited to toy datasets, and simulated astrophysics dataset. It would be good to observe the results with real data.
- It would help to compare OT not just with MSE, but also with other common domain-adaptation approaches (like adversarial methods or prototype-based matching).

**Score:**

4

**Topic Fit:**

3